# Towards Stabilizing Batch Statistics in Backward Propagation of Batch Normalization

**Junjie Yan**[1,2]*, **Ruosi Wan**[3]*, **Xiangyu Zhang**[3]†, **Wei Zhang**[1,2], **Yichen Wei**[3], **Jian Sun**[3]
[1] Shanghai Key Laboratory of Intelligent Information Processing
[2] School of Computer Science, Fudan University
[3] Megvii Technology.
{jjyan17, weizh}@fudan.edu.cn,
{wanruosi, zhangxiangyu, weiyichen, sunjian}@megvii.com.

## Abstract

Batch Normalization (BN) is one of the most widely used techniques in Deep Learning field. But its performance can awfully degrade with insufficient batch size. This weakness limits the usage of BN on many computer vision tasks like detection or segmentation, where batch size is usually small due to the constraint of memory consumption. Therefore many modified normalization techniques have been proposed, which either fail to restore the performance of BN completely, or have to introduce additional nonlinear operations in inference procedure and increase huge consumption. In this paper, we reveal that there are two extra batch statistics involved in backward propagation of BN, on which has never been well discussed before. The extra batch statistics associated with gradients also can severely affect the training of deep neural network. Based on our analysis, we propose a novel normalization method, named Moving Average Batch Normalization (MABN). MABN can completely restore the performance of vanilla BN in small batch cases, without introducing any additional nonlinear operations in inference procedure. We prove the benefits of MABN by both theoretical analysis and experiments. Our experiments demonstrate the effectiveness of MABN in multiple computer vision tasks including ImageNet and COCO. The code has been released in `https://github.com/megvii-model/MABN`.

## 1 Introduction

Batch Normalization (BN) (Ioffe & Szegedy, 2015) is one of the most popular techniques for training neural networks. It has been widely proven effective in many applications, and become the indispensable part of many state of the art deep models.

Despite the success of BN, it's still challenging to utilize BN when batch size is extremely small[1]. The batch statistics with small batch size are highly unstable, leading to slow convergence during training and bad performance during inference. For example, in detection or segmentation tasks, the batch size is often limited to 1 or 2 per GPU due to the requirement of high resolution inputs or complex structure of the model. Directly computing batch statistics without any modification on each GPU will make performance of the model severely degrade.

To address such issues, many modified normalization methods have been proposed. They can be roughly divided into two categories: some of them try to improve vanilla BN by correcting batch statistics (Ioffe, 2017; Singh & Shrivastava, 2019), but they all fail to completely restore the performance of vanilla BN; Other methods get over the instability of BN by using instance-level normalization (Ulyanov et al., 2016; Ba et al., 2016; Wu & He, 2018), therefore models can avoid the affect

---

*Equal Contribution. Work was done when Junjie Yan was an intern at Megvii Technology.
†Corresponding author.
[1]In the context of this paper, we use "batch size/normalization batch size" to refer the number of samples used to compute statistics unless otherwise stated. We use "gradient batch size" to refer the number of samples used to update weights.

of batch statistics. This type of methods can restore the performance in small batch cases to some extent. However, instance-level normalization hardly meet industrial or commercial needs so far, for this type of methods have to compute instance-level statistics both in training and inference, which will introduce additional nonlinear operations in inference procedure and dramatically increase consumption Shao et al. (2019). While vanilla BN uses the statistics computed over the whole training data instead of batch of samples when training finished. Thus BN is a linear operator and can be merged with convolution layer during inference procedure. Figure 1(a) shows with ResNet-50 (He et al., 2016), instance-level normalization almost double the inference time compared with vanilla BN. Therefore, it's a tough but necessary task to restore the performance of BN in small batch training without introducing any nonlinear operations in inference procedure.

In this paper, we first analysis the formulation of vanilla BN, revealing there are actually not only 2 but 4 batch statistics involved in normalization during forward propagation (FP) as well as backward propagation (BP). The additional 2 batch statistics involved in BP are associated with gradients of the model, and have never been well discussed before. They play an important role in regularizing gradients of the model during BP. In our experiments (see Figure 2), variance of the batch statistics associated with gradients in BP, due to small batch size, is even larger than that of the widely-known batch statistics (mean, variance of feature maps). We believe the instability of batch statistics associated with gradients is one of the key reason why BN performs poorly in small batch cases.

Based on our analysis, we propose a novel normalization method named *Moving Average Batch Normalization (MABN)*. MABN can completely get over small batch issues without introducing any nonlinear manipulation in inference procedure. The core idea of MABN is to replace batch statistics with moving average statistics. We substitute batch statistics involved in BP and FP with different type of moving average statistics respectively, and theoretical analysis is given to prove the benefits. However, we observed directly using moving average statistics as substitutes for batch statistics can't make training converge in practice. We think the failure takes place due to the occasional large gradients during training, which has been mentioned in Ioffe (2017). To avoid training collapse, we modified the vanilla normalization form by reducing the number of batch statistics, centralizing the weights of convolution kernels, and utilizing renormalizing strategy. We also theoretically prove the modified normalization form is more stable than vanilla form.

MABN shows its effectiveness in multiple vision public datasets and tasks, including ImageNet (Russakovsky et al., 2015), COCO (Lin et al., 2014). All results of experiments show MABN with small batch size (1 or 2) can achieve comparable performance as BN with regular batch size (see Figure 1(b)). Besides, it has same inference consumption as vanilla BN (see Figure 1(a)). We also conducted sufficient ablation experiments to verify the effectiveness of MABN further.

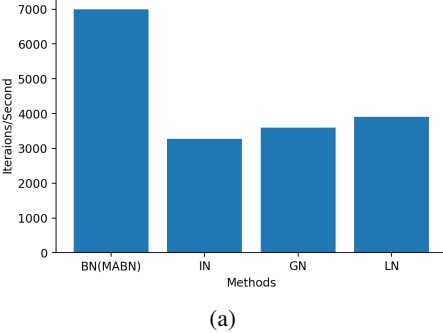
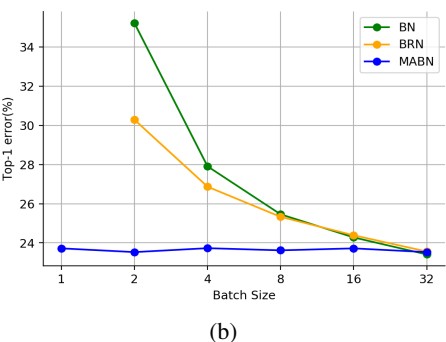

(a)                                           (b)

Figure 1: (a) Throughout (iterations per second) in inference procedure using different Normalization methods. The implementation details can be seen in appendix B.2. (b)ImageNet classification validation error vs. batch sizes.

## 2   RELATED WORK

Batch normalization (BN) (Ioffe & Szegedy, 2015) normalizes the internal feature maps of deep neural network using channel-wise statistics (mean, standard deviation) along batch dimension. It has

been widely proven effectively in most of tasks. But the vanilla BN heavily relies on sufficient batch size in practice. To restore the performance of BN in small batch cases, many normalization techniques have been proposed: Batch Renormalization (BRN) (Ioffe, 2017) introduces renormalizing parameters in BN to correct the batch statistics during training, where the renormalizing parameters are computed using moving average statistics; Unlike BRN, EvalNorm (Singh & Shrivastava, 2019) corrects the batch statistics during inference procedure. Both BRN and EvalNorm can restore the performance of BN to some extent, but they all fail to get over small batch issues completely. Instance Normalization (IN) (Ulyanov et al., 2016), Layer Normalization (LN) (Ba et al., 2016), and Group normalization (GN) (Wu & He, 2018) all try to avoid the effect of batch size by utilizing instance level statistics. IN uses channel-wise statistics per instance instead of per batch, while LN uses instance-level statistics along channel dimension. But IN and LN shows no superiority to vanilla BN in most of cases. GN divides all channels in predefined groups, and uses group-wise statistics per instance. It can restore the performance of vanilla BN very well in classification and detection tasks. But it have to introduce extra nonlinear manipulations in inference procedure and severely increase inference consumption, as we have pointed out in Section 1. SyncBN (Peng et al., 2018) handle the small batch issues by computing the mean and variance across multiple GPUs. This method doesn't essentially solve the problem, and requires a lot of resource. Online Normalization Chiley et al. (2019) modifies BP by using moving average statistics, so they can set batch size as 1 without degradation of performance, but Online Normalization still have to use instance-level normalization to cooperate with modification in BP, so its inference efficiency is much lower than original BN.

Apart from operating on feature maps, some works exploit to normalize the weights of convolution: Weight Standardization (Qiao et al., 2019) centralizes weight at first before divides weights by its standard deviation. It still has to combine with GN to handle small batch cases.

# 3    STATISTICS IN BATCH NORMALIZATION

## 3.1    REVIEW OF BATCH NORMALIZATION

First of all, let's review the formulation of batch Normalization (Ioffe & Szegedy, 2015): assume the input of a BN layer is denoted as $\boldsymbol{X} \in \mathbb{R}^{B \times p}$, where $B$ denotes the batch size, $p$ denotes number of features. In training procedure, the normalized feature maps $\boldsymbol{Y}$ at iteration $t$ is computed as:

$$\boldsymbol{Y} = \frac{\boldsymbol{X} - \mu_{\mathcal{B}_t}}{\sigma_{\mathcal{B}_t}}, \tag{1}$$

where batch statistics $\mu_{\mathcal{B}_t}$ and $\sigma^2_{\mathcal{B}_t}$ are the sample mean and sample variance computed over the batch of samples $\mathcal{B}_t$ at iteration $t$:

$$\mu_{\mathcal{B}_t} = \frac{1}{B} \sum_b \boldsymbol{X}_{b,:}, \quad \sigma^2_{\mathcal{B}_t} = \frac{1}{B} \sum_b (\boldsymbol{X}_{b,:} - \mu_{\mathcal{B}_t})^2. \tag{2}$$

Besides, a pair of parameters $\gamma$, $\beta$ are used to scale and shift normalized value $\boldsymbol{Y}$:

$$\boldsymbol{Z} = \boldsymbol{Y}\gamma + \beta. \tag{3}$$

The scaling and shifting part is added in all normalization form by default, and will be omitted in the following discussion for simplicity.

As Ioffe & Szegedy (2015) demonstrated, the batch statistics $\mu_{\mathcal{B}_t}, \sigma^2_{\mathcal{B}_t}$ are both involved in backward propagation (BP). We can derive the formulation of BP in BN as follows: let $\mathcal{L}$ denote the loss, $\Theta_t$ denote the set of the whole learnable parameters of the model at iteration $t$. Given the partial gradients $\frac{\partial \mathcal{L}}{\partial \boldsymbol{Y}}\Big|_{\Theta_t, \mathcal{B}_t}$, the partial gradients $\frac{\partial \mathcal{L}}{\partial \boldsymbol{X}}\Big|_{\Theta_t, \mathcal{B}_t}$ is computed as

$$\frac{\partial \mathcal{L}}{\partial \boldsymbol{X}}\Big|_{\Theta_t, \mathcal{B}_t} = \frac{1}{\sigma_{\mathcal{B}_t}} \left( \frac{\partial \mathcal{L}}{\partial \boldsymbol{Y}}\Big|_{\Theta_t, \mathcal{B}_t} - g_{\mathcal{B}_t} - \boldsymbol{Y} \cdot \Psi_{\mathcal{B}_t} \right) \tag{4}$$

where $\cdot$ denotes element-wise production, $g_{\mathcal{B}_t}$ and $\Psi_{\mathcal{B}_t}$ are computed as

$$g_{\mathcal{B}_t} = \frac{1}{B} \sum_b \frac{\partial \mathcal{L}}{\partial \boldsymbol{Y}_{b,:}}\Big|_{\Theta_t, \mathcal{B}_t}, \quad \Psi_{\mathcal{B}_t} = \frac{1}{B} \sum_b \boldsymbol{Y}_{b,:} \cdot \frac{\partial \mathcal{L}}{\partial \boldsymbol{Y}_{b,:}}\Big|_{\Theta_t, \mathcal{B}_t}, \tag{5}$$

It can be seen from (5) that $g_{\mathcal{B}_t}$ and $\Psi_{\mathcal{B}_t}$ are also batch statistics involved in BN during BP. But they have never been well discussed before.

## 3.2 INSTABILITY OF BATCH STATISTICS

According to Ioffe & Szegedy (2015), the ideal normalization is to normalize feature maps $\boldsymbol{X}$ using expectation and variance computed over the whole training data set:

$$\boldsymbol{Y} = \frac{\boldsymbol{X} - \mathbb{E}\boldsymbol{X}}{\sqrt{Var[\boldsymbol{X}]}}. \tag{6}$$

But it's impractical when using stochastic optimization. Therefore, Ioffe & Szegedy (2015) uses mini-batches in stochastic gradient training, each mini-batch produces estimates the mean and variance of each activation. Such simplification makes it possible to involve mean and variance in BP. From the derivation in section 3.1, we can see batch statistics $\mu_{\mathcal{B}_t}$, $\sigma^2_{\mathcal{B}_t}$ are the Monte Carlo (MC) estimators of population statistics $\mathbb{E}[\boldsymbol{X}|\boldsymbol{\Theta}_t]$, $Var[\boldsymbol{X}|\boldsymbol{\Theta}_t]$ respectively at iteration $t$. Similarly, batch statistics $g_{\mathcal{B}_t}$, $\Psi_{\mathcal{B}_t}$ are MC estimators of population statistics $\mathbb{E}[\frac{\partial \mathcal{L}}{\partial \boldsymbol{Y}_{b,:}}|\boldsymbol{\Theta}_t]$, $\mathbb{E}[\boldsymbol{Y}_{b,:} \cdot \frac{\partial \mathcal{L}}{\partial \boldsymbol{Y}_{b,:}}|\boldsymbol{\Theta}_t]$ at iteration $t$. $\mathbb{E}[\frac{\partial \mathcal{L}}{\partial \boldsymbol{Y}_{b,:}}|\boldsymbol{\Theta}_t]$, $\mathbb{E}[\boldsymbol{Y}_{b,:} \cdot \frac{\partial \mathcal{L}}{\partial \boldsymbol{Y}_{b,:}}|\boldsymbol{\Theta}_t]$ are computed over the whole data set. They contain the information how the mean and the variance of population will change as model updates, so they play an important role to make trade off between the change of individual sample and population. Therefore, it's crucial to estimate the population statistics precisely, in order to regularize the gradients of the model properly as weights update.

It's well known the variance of MC estimator is inversely proportional to the number of samples, hence the variance of batch statistics dramatically increases when batch size is small. Figure 2 shows the change of batch statistics from a specific normalization layer of ResNet-50 during training on ImageNet. Regular batch statistics (orange line) are regarded as a good approximation for population statistics. We can see small batch statistics (blue line) are highly unstable, and contains notable error compared with regular batch statistics during training. In fact, the bias of $g_{\mathcal{B}_t}$ and $\Psi_{\mathcal{B}_t}$ in BP is more serious than that of $\mu_{\mathcal{B}_t}$ and $\sigma^2_{\mathcal{B}_t}$ (see Figure 2(c), 2(d)). The instability of small batch statistics can worsen the capacity of the models in two aspects: firstly the instability of small batch statistics will make training unstable, resulting in slow convergence; Secondly the instability of small batch can produce huge difference between batch statistics and population statistics. Since the model is trained using batch statistics while evaluated using population statistics, the difference between batch statistics and population statistics will cause inconsistency between training and inference procedure, leading to bad performance of the model on evaluation data.

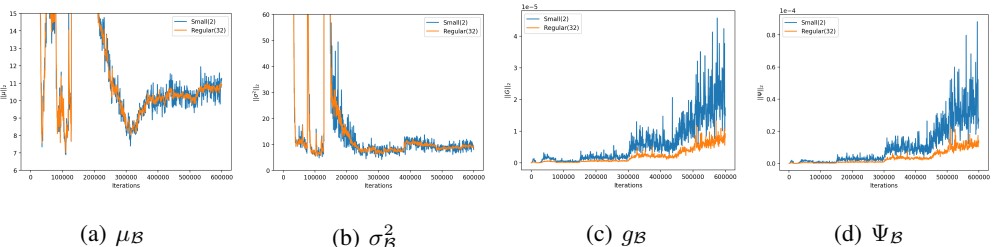

(a) $\mu_{\mathcal{B}}$      (b) $\sigma^2_{\mathcal{B}}$      (c) $g_{\mathcal{B}}$      (d) $\Psi_{\mathcal{B}}$

Figure 2: Plot of batch statistics from layer1.0.bn1 in ResNet-50 during training. The formulation of these batch statistics ($\mu_{\mathcal{B}}, \sigma^2_{\mathcal{B}}, g_{\mathcal{B}}, \Psi_{\mathcal{B}}$) have been shown in Section 3.1. Blue line represents the small batch statistic ($|\mathcal{B}| = 2$) to compute, while orange line represents the regular batch statistics($|\mathcal{B}| = 32$). The x-axis represents the iterations, while the y-axis represents the $l^2$ norm of these statistics in each figures. Notice the mean of $g$ and $\Psi$ is close to zero, hence $l^2$ norm of $g_{\mathcal{B}}$ and $\Psi_{\mathcal{B}}$ essentially represent their standard deviation.

## 4 MOVING AVERAGE BATCH NORMALIZATION

Based on the discussion in Section 3.2, the key to restore the performance of BN is to solve the instability of small batch statistics. Therefore we considered two ways to handle the instability of

small batch statistics: using moving average statistics to estimate population statistics, and reducing the number of statistics by modifying the formulation of normalization.

## 4.1 SUBSTITUTE BATCH STATISTICS BY MOVING AVERAGE STATISTICS.

Moving average statistics seem to be a suitable substitute for batch statistics to estimate population statistics when batch is small. We consider two types of moving average statistics: simple moving average statistics (SMAS)[2] and exponential moving average statistics (EMAS)[3]. The following theorem shows under mild conditions, SMAS and EMAS are more stable than batch statistics:

**Theorem 1** *Assume there exists a sequence of random variable (r.v.) $\{\xi_t\}_{t=1}^{\infty}$, which are independent, uniformly bounded, i.e. $\forall t, |\xi_t| < C$, and have uniformly bounded density. Define:*

$$S_t = \frac{1}{m} \sum_{i=t-m+1}^{t} \xi_i, \quad E_t = (1-\alpha) \sum_{i=1}^{t} \alpha^{t-i} \xi_i, \tag{7}$$

*where $m \in \mathbb{R}^+$. If the sequence $\{\xi_t\}_{t=1}^{\infty}$ satisfies*

$$\exists \xi, \forall \epsilon \in \mathbb{R}, \lim_{t \to \infty} P(\xi_t \leq \epsilon) = P(\xi \leq \epsilon), \tag{8}$$

*then we have*

$$\mathbb{E}(E_t) = \mathbb{E}(\xi) + o(1), \quad Var(E_t) = \frac{(1-\alpha^{2t})(1-\alpha)}{1+\alpha} Var(\xi) + o(1); \tag{9}$$

*If the sequence $\{\xi_t\}_{t=1}^{\infty}$ satisfies*

$$\lim_{t \to \infty} \sup_{\lambda} |P(\xi_{t-1} < \lambda) - P(\xi_t < \lambda)| = 0, \tag{10}$$

*then we have*

$$\mathbb{E}(S_t) = \mathbb{E}(\xi_t) + o(1), \quad Var(S_t) = \frac{Var(\xi_t)}{m} + o(1); \tag{11}$$

The proof of theorem 1 can be seen in appendix A.1. Theorem 1 not only proves moving average statistics have lower variance compared with batch statistics, but also reveals that with large momentum $\alpha$, EMAS is better than SMAS with lower variance. However, using SMAS and EMAS request different conditions: Condition (8) means the sequence of the given statistics need to weakly converge to a specific random variable. For $\{\mu_{\mathcal{B}_t}\}_{t=1}^{\infty}, \{\sigma_{\mathcal{B}_t}^2\}_{t=1}^{\infty}$, they converge to the "final" batch statistics $\mu_{\mathcal{B}}, \sigma_{\mathcal{B}}^2$ (when training finished), hence condition (8) is satisfied, EMAS can be applied to replace $\{\mu_{\mathcal{B}_t}\}_{t=1}^{\infty}, \{\sigma_{\mathcal{B}_t}^2\}_{t=1}^{\infty}$; Unfortunately $\{g_{\mathcal{B}_t}\}_{t=1}^{\infty}, \{\Psi_{\mathcal{B}_t}\}_{t=1}^{\infty}$ don't share the same property, EMAS is not suitable to take replace of $\{g_{\mathcal{B}_t}\}_{t=1}^{\infty}, \{\Psi_{\mathcal{B}_t}\}_{t=1}^{\infty}$. However, under the assumption that learning rate is extremely small, the difference between the distribution of $\xi_{t-1}$ and $\xi_t$ is tiny, thus condition (10) is satisfied, we can use SMAS to replace $\{g_{\mathcal{B}_t}\}_{t=1}^{\infty}, \{\Psi_{\mathcal{B}_t}\}_{t=1}^{\infty}$. In a word, we can use EMAS $\hat{\mu}_t, \hat{\sigma}_t^2$ to replace $\mu_{\mathcal{B}_t}, \sigma_{\mathcal{B}_t}^2$, and use SMAS $\bar{g}_t, \bar{\Psi}_t$ to replace $g_{\mathcal{B}_t}, \Psi_{\mathcal{B}_t}$ in (1) and (4), where

$$\hat{\mu}_t = \alpha \hat{\mu}_{t-1} + (1-\alpha)\mu_{\mathcal{B}_t}, \quad \hat{\sigma}_t^2 = \alpha \hat{\sigma}_{t-1}^2 + (1-\alpha)\sigma_{\mathcal{B}_t}^2, \tag{12}$$

$$\bar{g}_t = \frac{1}{m} \sum_{s=1}^{m} g_{\mathcal{B}_{t-m+s}}, \quad \bar{\Psi}_t = \frac{1}{m} \sum_{s=1}^{m} \Psi_{\mathcal{B}_{t-m+s}}. \tag{13}$$

Notice neither of SMAS and EMAS is the unbiased substitute for batch statistics, but the bias can be extremely small comparing with expectation and variance of batch statistics, which is proven by equation 11 in theorem 1, our experiments also prove the effectiveness of moving average statistics as substitutes for small batch statistics (see Figure 3, 4 in appendix B.1).

---

[2]The exponential moving average (EMA) for a series $\{Y_t\}_{t=1}^{\infty}$ is calculated as: $S_t = \alpha \cdot Y_t + (1-\alpha) \cdot S_{t-1}$.

[3]The simple moving average (SMA) for a series $\{Y_t\}_{t=1}^{\infty}$ is calculated as: $S_t = \frac{\sum_{s=t-M+1}^{t} Y_s}{M}$.

**Relation to Batch Renormalization** Essentially, Batch Renormalization (BRN) (Ioffe, 2017) replaces batch statistics $\mu_{\mathcal{B}_t}$, $\sigma^2_{\mathcal{B}_t}$ with EMAS $\hat{\mu}_t$, $\hat{\sigma}^2_t$ both in FP (1) and BP (4). The formulation of BRN during training is written as:

$$\boldsymbol{Y} = \frac{\boldsymbol{X} - \mu_{\mathcal{B}_t}}{\sigma_{\mathcal{B}_t}}, \quad \hat{\boldsymbol{Y}} = r \cdot \boldsymbol{Y} + d \tag{14}$$

where $r = clip_{[1/\lambda, \lambda]}(\frac{\sigma_{\mathcal{B}_t}}{\hat{\sigma}_t})$, $d = clip_{[-d,d]}(\frac{\mu_{\mathcal{B}_t} - \hat{\mu}_t}{\hat{\sigma}_t})$. Based on our analysis, BRN successfully eliminates the effect of small batch statistics $\mu_{\mathcal{B}_t}$ and $\sigma^2_{\mathcal{B}_t}$ by EMAS, but the small batch statistics associated with gradients $g_{\mathcal{B}_t}$ and $\Psi_{\mathcal{B}_t}$ remains during backward propagation, preventing BRN from completely restoring the performance of vanilla BN.

## 4.2 Stabilizing Normalization by reducing the number of Statistics

To further stabilize training procedure in small batch cases, we consider normalizing feature maps $\boldsymbol{X}$ using $\mathbb{E}\boldsymbol{X}^2$ instead of $\mathbb{E}\boldsymbol{X}$ and $Var(\boldsymbol{X})$. The formulation of normalization is modified as:

$$\boldsymbol{Y} = \frac{\boldsymbol{X}}{\chi_{\mathcal{B}_t}}, \quad \boldsymbol{Z} = \boldsymbol{Y} \cdot \gamma + \beta, \tag{15}$$

where $\chi^2_{\mathcal{B}_t} = \frac{1}{B} \sum_b \boldsymbol{X}^2_{b,:}$. Given $\frac{\partial \mathcal{L}}{\partial \boldsymbol{Y}}$, the backward propagation is:

$$\left.\frac{\partial \mathcal{L}}{\partial \boldsymbol{X}}\right|_{\Theta_t, \mathcal{B}_t} = \frac{1}{\chi_{\mathcal{B}_t}} \left(\left.\frac{\partial \mathcal{L}}{\partial \boldsymbol{Y}}\right|_{\Theta_t, \mathcal{B}_t} - \boldsymbol{Y} \cdot \Psi_{\mathcal{B}_t}\right). \tag{16}$$

The benefits of the modification seems obvious: there's only two batch statistics left during FP and BP, which will introduce less instability into the normalization layer compared with vanilla normalizing form. In fact we can theoretically prove the benefits of the modification by following theorem:

**Theorem 2** *If the following assumptions hold:*

1. *$Var[\hat{\sigma}] = o(1)$, $Var[\hat{\chi}] = o(1)$;*

2. *$Cov(\{\frac{\partial \mathcal{L}}{\partial y}, y\}, \{g_{\mathcal{B}}, \Psi_{\mathcal{B}}\}) = o(1)$;*

3. *$\mathbb{E}y = o(1)$;*

*Then we have:*

$$Var\left[\left.\frac{\partial \mathcal{L}}{\partial x}\right|_{modified}\right] \leq Var\left[\left.\frac{\partial \mathcal{L}}{\partial x}\right|_{vanilla}\right] - \frac{Var[g_{\mathcal{B}}]}{\hat{\sigma}^2} \tag{17}$$

The proof can be seen in appendix A.2. According to (17), $Var[\partial \mathcal{L}/\partial \boldsymbol{X}|_{vanilla}]$ is larger than that of $Var[\partial \mathcal{L}/\partial \boldsymbol{X}|_{modified}]$, the gap is at least $Var[g_{\mathcal{B}}]/\hat{\sigma}^2$, which mainly caused by the variance of $g_{\mathcal{B}}/\hat{\sigma}$. So the modification essentially reduces the variance of the gradient by eliminating the batch statistics $g_{\mathcal{B}}$ during BP. Since $g_{\mathcal{B}_t}$ is a Monte Carlo estimator, the gap is inversely proportional to batch size. This can also explain why the improvement of modification is significant in small batch cases, but modified BN shows no superiority to vanilla BN within sufficient batch size (see ablation study in section 5.1).

**Centralizing weights of convolution kernel** Notice theorem 2 relies on assumption 3. The vanilla normalization naturally satisfies $\mathbb{E}y = 0$ by centralizing feature maps, but the modified normalization doesn't necessarily satisfy assumption 3. To deal with that, inspired by Qiao et al. (2019), we find centralizing weights $\boldsymbol{W} \in \mathbb{R}^{q \times p}$ of convolution kernels, named as Weight Centralization (WC) can be a compensation for the absence of centralizing feature maps in practice:

$$\bar{\boldsymbol{W}} = \frac{1}{p} \sum_i \boldsymbol{W}_{:i}, \quad \boldsymbol{X}_{output} = (\boldsymbol{W} - \bar{\boldsymbol{W}})\boldsymbol{X}_{input}, \tag{18}$$

where $\boldsymbol{X}_{input}$, $\boldsymbol{X}_{output}$ are the input and output of the convolution layer respectively. We conduct further ablation study to clarify the effectiveness of WC (see Table 4 in appendix B.2). It shows that WC has little benefits to vanilla normalization, but it can significantly improve the performance of modified normalization. We emphasize that weight centralization is only a practical remedy for the absence of centralizing feature maps. The theoretical analysis remains as a future work.

**Clipping and renormalizing strategy.** In practice, we find directly substituting batch statistics by moving average statistics in normalization layer will meet collapse during training. Therefore we take use of the clipping and renormalizing strategy from BRN (Ioffe, 2017).

All in all, the formulation of proposed method MABN is:

$$\boldsymbol{Y} = \frac{\boldsymbol{X}}{\bar{\chi}_t}, \quad \hat{\boldsymbol{Y}} = r \cdot \boldsymbol{Y} \tag{19}$$

$$\frac{\partial \mathcal{L}}{\partial \boldsymbol{Y}}\bigg|_{\Theta_t, \mathcal{B}_t} = r \cdot \frac{\partial \mathcal{L}}{\partial \hat{\boldsymbol{Y}}}\bigg|_{\Theta_t, \mathcal{B}_t}, \quad \frac{\partial \mathcal{L}}{\partial \boldsymbol{X}}\bigg|_{\Theta_t, \mathcal{B}_t} = \frac{1}{\bar{\chi}_t}(\frac{\partial \mathcal{L}}{\partial \boldsymbol{Y}}\bigg|_{\Theta_t, \mathcal{B}_t} - \boldsymbol{Y} \odot \bar{\Psi}_t) \tag{20}$$

where the EMAS $\hat{\chi}_t$ is computed as $\hat{\chi}_t = \alpha\hat{\chi}_{t-1} + (1 - \alpha)\chi_{\mathcal{B}_t}$, SMAS $\bar{\chi}_t$ is defined as $\bar{\chi}_t^2 = \frac{1}{m}\sum_{s=1}^{m}\chi_{\mathcal{B}_{t-m+s}}^2$, SMAS $\bar{\Psi}_t$ is defined as (13). The renormalizing parameter is set as $r = clip_{[1/\lambda, \lambda]}(\frac{\bar{\chi}_t}{\hat{\chi}_t})$.

## 5 EXPERIMENTS

This section presents main results of MABN on ImageNet (Russakovsky et al., 2015), COCO (Lin et al., 2014). Further experiment results on ImangeNet, COCO and Cityscapes (Cordts et al., 2016) can be seen in appendix B.2, B.3, B.4 resepectively. We also evaluate the computational overhead and memory footprint of MABN, the results is shown in appendix B.5.

### 5.1 IMAGE CLASSIFICATION IN IMAGENET

We evaluate the proposed method on ImageNet (Russakovsky et al., 2015) classification datatsets with 1000 classes. All classification experiments are conducted with ResNet-50 (He et al., 2016). More implementation details can be found in the appendix B.2.

| | BN (Regular) | BN (Small) | BRN (Small) | MABN (Small, $m = 16$) |
|---|---|---|---|---|
| val error | **23**.41 | 35.22 | 30.29 | 23.58 |
| Δ (vs BN(Regular)) | - | 11.81 | 6.88 | **0.17** |

Table 1: **Comparison of top-1 error rate (%) of ResNet-50 on ImageNet Classification.** The gradient batch size is 32 per GPU. **Regular** means normalization batch size is 32, while **Small** means normalization batch size is 2.

**Comparison with other normalization methods.** Our baseline is BN using small ($|\mathcal{B}| = 2$) or regular ($|\mathcal{B}| = 32$) batch size, and BRN (Ioffe, 2017) with small batch size. We don't present the performance of instance-level normalization counterpart on ImageNet, because they are not linear-type method during inference time, and they also failed to restore the performance of BN (over +0.5%), according to Wu & He (2018). Table 1 shows vanilla BN with small batch size can severely worsen the performance of the model(+11.81%); BRN (Ioffe, 2017) alleviates the issue to some extent, but there's still remaining far from complete recovery(+6.88%); While MABN almost completely restore the performance of vanilla BN(+0.17%).

We also compared the performance of BN, BRN and MABN when varying the batch size (see Figure 1(b)). BN and BRN are heavily relies on the batch size of training, though BRN performs better than

vanilla BN. MABN can always retain the best capacity of ResNet-50, regardless of batch size during training.

| Experiment Number | Vanilla Normalization | Modified Normalization | EMAS in FP | SMAS in BP | Top-1 Error (%) |
|---|---|---|---|---|---|
| ① | ✓ | | | | 23.41 (BN, regular) |
| ② | | ✓ | | | 23.53 (regular) |
| ③ | ✓ | | | | 35.22 (BN) |
| ④ | ✓ | | ✓ | | 30.29 (BRN) |
| ⑤ | ✓ | | ✓ | ✓ | - |
| ⑥ | | ✓ | | | 29.68 |
| ⑦ | | ✓ | ✓ | | 27.03 |
| ⑧ | | ✓ | ✓ | ✓ | 23.58 (MABN) |

Table 2: **Ablation study on ImageNet Classification with ResNet-50.** The normalization batch size is 2 in all experiments otherwise stated. The memory size is 16 and momentum is 0.98 when using SMAS, otherwise the momentum is 0.9. "-" means the training can't converge.

**Ablation study on ImageNet.**  We conduct ablation experiments on ImageNet to clarify the contribution of each part of MABN (see table 2). With vanilla normalization form, replacing batch statistics in FP with EMAS (as BRN) will restore the performance to some extents($-4.93\%$, comparing ③ and ④), but there's still a huge gap ($+6.88\%$, comparing ① and ④) from complete restore. Directly using SMAS in BP with BRN will meet collapse during training (⑤), no matter how we tuned hyperparameters. We think it's due to the instability of vanilla normalization structure in small cases, so we modify the formulation of normalization shown in section 4.2. The modified normalization even slightly outperforms BRN in small batch cases (comparing ④ and ⑥). However, modified normalization shows no superiority to vanilla form (comparing ① and ②), which can be interpreted by the result of theorem 2. With EMAS in FP, modified normalization significantly reduces the error rate further (comparing ⑥ and ⑦), but still fail to restore the performance completely ($+3.62\%$, comparing ① and ⑦). Applying SMAS in BP finally fills the rest of gap, almost completely restore the performance of vanilla BN in small batch cases ($+0.17$ ,comparing ① and ⑧).

## 5.2   Detection and Segmentation in COCO from scratch

We conduct experiments on Mask R-CNN (He et al., 2017) benchmark using a Feature Pyramid Network(FPN) (Lin et al., 2017a) following the basic setting in He et al. (2017). We train the networks from scratch (He et al., 2018) for $2\times$ times. Only the backbone contains normalization layers. More implementation details and experiment results can be seen in the appendix B.3.

| | $AP^{bbox}$ | $AP^{bbox}_{50}$ | $AP^{bbox}_{75}$ | $AP^{mask}$ | $AP^{mask}_{50}$ | $AP^{mask}_{75}$ |
|---|---|---|---|---|---|---|
| BN | **30.41** | 48.47 | 32.70 | **27.91** | 45.79 | 29.33 |
| BRN | **31.93** | 50.95 | 34.48 | **29.16** | 48.16 | 30.69 |
| SyncBN | **34.81** | 55.18 | 37.69 | **31.69** | 51.86 | 33.68 |
| MABN | **34.85** | 54.97 | 38.00 | **31.61** | 51.88 | 33.64 |

Table 3: Comparison of Average Precision(AP) of Mask-RCNN on COCO Detection and Segmentation. The gradients batch size is 16. The normalization batch size of SyncBN is 16, while that of BN, BRN and MABN are both 2. The momentum of BRN and MABN are both 0.98, while the momentum of BN and SyncBN are both 0.9. The buffer size ($m$) is 16).

Table 3 shows the result of MABN compared with vanilla BN, BRN and SyncBN (Peng et al., 2018). It can be seen that MABN outperforms vanilla BN and BRN by a clear margin and get comparable performance with SyncBN. Quite different from Imagenet experiments, we update the parameters every single batch (with $B_{norm} = 2$). With such a complex pipeline, MABN still achieves a comparable performance as SyncBN.

## 6 CONCLUSION

This paper reveals the existence of the batch statistics $g_B$ and $\Psi_B$ involved in backward propagation of BN, and analysis their influence to training process. This discovery provides a new perspective to understand why BN always fails in small batch cases. Based on our analysis, we propose MABN to deal with small batch training problem. MABN can completely restore the performance of vanilla BN in small batch cases, and is extraordinarily efficient compared with its counterpart like GN. Our experiments on multiple computer vision tasks (classification, detection, segmentation) have shown the remarkable performance of MABN.

## ACKNOWLEDGEMENT

This research was partially supported by National Key RD Program of China (No. 2017YFA0700800), Beijing Academy of Artificial Intelligence (BAAI), and NSFC under Grant No. 61473091.

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

# A  SKETCH OF PROOF

## A.1  PROOF OF THEOREM 1

If the condition (8) is satisfied, i.e. $\{\xi_t\}_{t=1}^{\infty}$ weakly converge to $\xi$. Since $\{\xi_t\}_{t=1}^{\infty}$ has uniformly bounded density, we have:

$$\lim_{t\to\infty}\mathbb{E}\xi_t = \mathbb{E}\xi \tag{21}$$

$$\lim_{t\to\infty}Var[\xi_t] = Var[\xi] \tag{22}$$

Since $\{\xi_t\}_{t=1}^{\infty}$ are independently, hence we have:

$$
\begin{aligned}
Var[E_t] &= Var[(1-\alpha)\sum_{i=1}^{t}\sum_{i=1}^{t}\alpha^{t-i}\xi_i] \\
&= (1-\alpha)^2\sum_{i=1}^{t}\alpha^{2(t-i)}Var[\xi_i] \\
&= (1-\alpha)^2\sum_{i=1}^{t}\alpha^{2(t-i)}Var[\xi] + (1-\alpha)^2\sum_{i=1}^{t}\alpha^{2(t-i)}(Var[\xi_i]-Var[\xi]) \\
&= \frac{(1-\alpha^{2t})(1-\alpha)}{1+\alpha}Var[\xi] + o(1)
\end{aligned} \tag{23}
$$

as $t\to\infty$. Hence (9) has been proven.

If the condition (10) is satisfied. Since $\{\xi_t\}_{t=1}^{\infty}$ is uniformly bounded, then $\exists C\in\mathbb{R}^+, \forall, |\xi_t| < C$. As $t\to\infty$, We have

$$
\begin{aligned}
\left|\mathbb{E}\xi_{t-1}-\mathbb{E}\xi_t\right| &= \left|\int_{x\in[-C,C]}xp_{t-1}(x)dx - \int_{x\in[-C,C]}xp_t(x)dx\right| \\
&= \left|\int_{x\in[-C,C]}x(p_{t-1}(x)-p_t(x))dx\right| \\
&= \left|x(F_{t-1}(x)-F_t(x))\big|_{-C}^{C} - \int_{x\in[-C,C]}(F_{t-1}(x)-F_t(x))dx\right| \\
&\leq \int_{x\in[-C,C]}|F_{t-1}(x)-F_t(x)|dx \\
&\leq 2C\cdot\sup_x|F_{t-1}(x)-F_t(x)| \\
&= 2C\cdot\sup_x|P(\xi_{t-1}<x)-P(\xi_t<x)| \\
&= o(1)
\end{aligned} \tag{24}
$$

Similarly, we have

$$
\begin{aligned}
\left|\mathbb{E}\xi_{t-1}^2-\mathbb{E}\xi_t^2\right| &= \left|\int_{x\in[-C,C]}x^2(p_{t-1}(x)-p_t(x))dx\right| \\
&= \left|x^2(F_{t-1}(x)-F_t(x))\big|_{-C}^{C} - \int_{x\in[-C,C]}2x(F_{t-1}(x)-F_t(x))dx\right| \\
&\leq \int_{x\in[-C,C]}2|x||F_{t-1}(x)-F_t(x)|dx \\
&\leq 4C^2\cdot\sup_x|F_{t-1}(x)-F_t(x)| \\
&= o(1)
\end{aligned} \tag{25}
$$

Therefore combining (24) and (25), we have

$$
\begin{aligned}
|Var[\xi_{t-1}]-Var[\xi_t]| &\leq |E\xi_{t-1}^2-E\xi_t^2| + |(E\xi_{t-1})^2-(E\xi_t)^2| \\
&= o(1)
\end{aligned} \tag{26}
$$

For a fixed memory size $m$, as $t \to \infty$, we have

$$
\begin{aligned}
Var(S_t) &= Var[\frac{1}{m} \sum_{i=0}^{m-1} \xi_{t-i}] \\
&= \frac{1}{m} \sum_{i=0}^{m-1} Var[\xi_{t-i}] \\
&= \frac{1}{m} \sum_{i=0}^{m-1} (Var[\xi_t] + o(1)) \\
&= Var[\xi_t] + o(1)
\end{aligned}
\tag{27}
$$

Therefore, (11) has been proven.

### A.2 PROOF OF THEOREM 2

Without loss of generality, given the backward propagation of two normalizing form of a single input $x$ with batch $\mathcal{B}$:

$$
\frac{\partial \mathcal{L}}{\partial x}\Big|_{vanilla} = \frac{1}{\hat{\sigma}}[\frac{\partial \mathcal{L}}{\partial y} - g_{\mathcal{B}} - y \cdot \Psi_{\mathcal{B}}], \quad \frac{\partial \mathcal{L}}{\partial x}\Big|_{modified} = \frac{1}{\hat{\chi}}[\frac{\partial \mathcal{L}}{\partial y} - y \cdot \Psi_{\mathcal{B}}],
\tag{28}
$$

where $g_{\mathcal{B}}$, $\Psi_{\mathcal{B}}$ are the batch statistics, and $\hat{\sigma}$, $\hat{\chi}$ are the EMAS, defined as before. We omitted the subscript $t$ for simplicity. Then the variance of partial gradients w.r.t. inputs $x$ is written as

$$
\begin{aligned}
Var[\frac{\partial \mathcal{L}}{\partial x}\Big|_{vanilla}] &= Var[\frac{1}{\hat{\sigma}}[\frac{\partial \mathcal{L}}{\partial y} - g_{\mathcal{B}} - y \cdot \Psi_{\mathcal{B}}]] \tag{29} \\
&= \frac{1}{\hat{\sigma}^2}\Big[Var[\frac{\partial \mathcal{L}}{\partial y} - y \cdot \Psi_{\mathcal{B}}] + Var[g_{\mathcal{B}}] + 2Cov[\frac{\partial \mathcal{L}}{\partial y} - y \cdot \Psi_{\mathcal{B}}, g_{\mathcal{B}}]\Big] \tag{30} \\
&= \frac{1}{\hat{\sigma}^2}\Big[Var[\frac{\partial \mathcal{L}}{\partial y} - y \cdot \Psi_{\mathcal{B}}] + Var[g_{\mathcal{B}}]]\Big] \tag{31} \\
&\geq \frac{1}{\hat{\chi}^2}Var[\frac{\partial \mathcal{L}}{\partial y} - y \cdot \Psi_{\mathcal{B}}] + \frac{Var[g_{\mathcal{B}}]}{\hat{\sigma}^2} \tag{32} \\
&= Var[\frac{\partial \mathcal{L}}{\partial x}\Big|_{modified}] + \frac{Var[g_{\mathcal{B}}]}{\hat{\sigma}^2} \tag{33}
\end{aligned}
$$

where (30) is satisfied due to assumption 1. The variance of $\hat{\sigma}$ is so small that $\hat{\sigma}$ can be regarded as a fixed number; (31) is satisfied because

$$
\begin{aligned}
Cov[\frac{\partial \mathcal{L}}{\partial y} - y \cdot \Psi_{\mathcal{B}}, g_{\mathcal{B}}] &= Cov[\frac{\partial \mathcal{L}}{\partial y}, g_{\mathcal{B}}] - Cov[y \cdot \Psi_{\mathcal{B}}, g_{\mathcal{B}}] \tag{34} \\
&= Cov[\frac{\partial \mathcal{L}}{\partial y}, g_{\mathcal{B}}] - \mathbb{E}[y\Psi_{\mathcal{B}}(g_{\mathcal{B}} - \mathbb{E}g\mathcal{B})] + \mathbb{E}[y\Psi_{\mathcal{B}}]\mathbb{E}[g_{\mathcal{B}} - \mathbb{E}g_{\mathcal{B}}] \tag{35}
\end{aligned}
$$

Due to assumption 2, the correlation between individual sample and batch statistics is close to 0, hence we have

$$
\begin{aligned}
Cov[\frac{\partial \mathcal{L}}{\partial y}, g_{\mathcal{B}}] &= 0 \tag{36} \\
\mathbb{E}[y\Psi_{\mathcal{B}}(g_{\mathcal{B}} - \mathbb{E}g\mathcal{B})] &= \mathbb{E}y\mathbb{E}[\Psi_{\mathcal{B}}(g_{\mathcal{B}} - \mathbb{E}g\mathcal{B})] \tag{37} \\
\mathbb{E}[y\Psi_{\mathcal{B}}] &= \mathbb{E}y\mathbb{E}\Psi_{\mathcal{B}} \tag{38}
\end{aligned}
$$

Besides, $\mathbb{E}y$ is close to 0 according to assumption 3, hence

$$
Cov[\frac{\partial \mathcal{L}}{\partial y} - y \cdot \Psi_{\mathcal{B}}, g_{\mathcal{B}}] = 0.
\tag{39}
$$

(32) is satisfied due to the definition of $\hat{\chi}$ and $\hat{\sigma}$, we have

$$
\hat{\chi}^2 = \hat{\sigma}^2 + \hat{\mu}^2.
\tag{40}
$$

Similar to $\hat{\sigma}$, the variance of $\hat{\chi}$ is also too small that $\hat{\chi}$ can be regarded as a fixed number due to assumption 1, so (33) is satisfied.

## B EXPERIMENTS

### B.1 STATISTICS ANALYSIS

We analyze the difference between small batch statistics ($|\mathcal{B}| = 2$) and regular batch statistics ($|\mathcal{B}| = 32$) with the modified formulation of normalization (15) shown in Section 4.2.

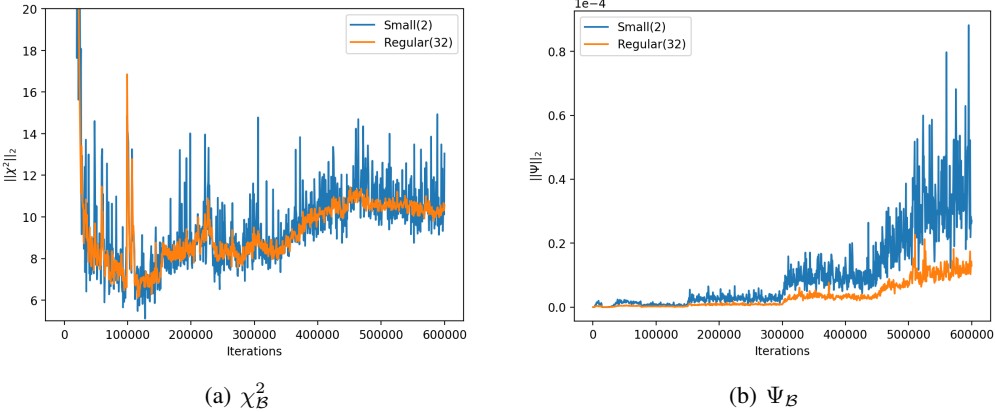

(a) $\chi^2_{\mathcal{B}}$          (b) $\Psi_{\mathcal{B}}$

Figure 3: Plot of batch statistics from layer1.0.bn1 in ResNet-50 with a modified structure during training. The formulation of these batch statistics ($\chi^2_{\mathcal{B}}$, $\Psi_{\mathcal{B}}$) is shown in section 4.2, 3.1 respectively. Blue line represents the small batch statistic ($|\mathcal{B}| = 2$), while orange line represents the regular batch statistics ($|\mathcal{B}| = 32$). We use the small batch statistics to update the network parameters.

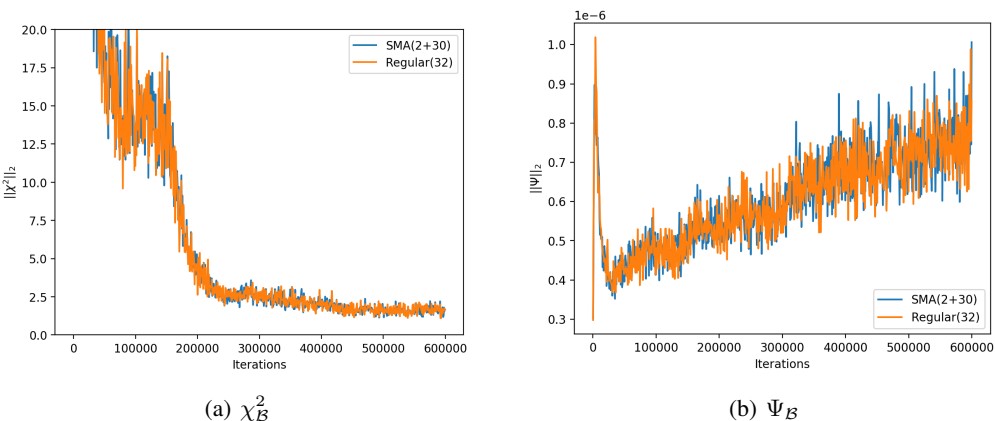

(a) $\chi^2_{\mathcal{B}}$          (b) $\Psi_{\mathcal{B}}$

Figure 4: Plot of batch statistics from layer1.0.bn1 in ResNet-50 with MABN. The formulation of these batch statistics ($\chi^2_{\mathcal{B}}$, $\Psi_{\mathcal{B}}$) is shown in section 4.2, 3.1 respectively. Blue line represents the SMA batch statistic(2+30), while orange line represents the regular batch statistics(32). We use the moving average batch statistics to update the network parameters.

Figure 3 illustrates the change of small batch statistics and regular batch statistics in FP and BP respectively. The variance of small batch statistics is much higher than the regular one. However, when we use SMAS as a approximation for regular batch statistics, the gap between SMAS and regular batch statistics is not obvious as shown in Figure 4.

## B.2 Experiments on ImageNet

**Implementation details.** All experiments on ImageNet are conducted across 8 GPUs. We train models with a gradient batch size of $B_g = 32$ images per GPU. To simulate small batch training, we split the samples on each GPU into $B_g/|\mathcal{B}|$ groups where $|\mathcal{B}|$ denotes the normalization batch size. The batch statistics are computed within each group individually.

All weights from convolutions are initialized as He et al. (2015). We use 1 to initialize all $\gamma$ and 0 to initialize all $\beta$ in normalization layers. We use a weight decay of $10^{-4}$ for all weight layers including $\gamma$ and $\beta$ (following Wu & He (2018)). We train $600,000$ iterations (approximately equal to 120 epoch when gradient batch size is 256) for all models, and divide the learning rate by 10 at $150,000$, $300,000$ and $450,000$ iterations. The data augmentation follows Gross & Wilber (2016). The models are evaluated by top-1 classification error on center crops of $224 \times 224$ pixels in the validation set. In vanilla BN or BRN, the momentum $\alpha = 0.9$, in MABN, the momentum $\alpha = 0.98$.

**Additional ablation studies.** Table 4 shows the additional ablation results. We test all possible combination of all three kinds of statistics (SMAS, EMAS, BS) in FP and BP. The experiments results strongly prove our theoretical analysis in section 4.3. Besides, we verify the necessity of centralizing weights with modified normalization form.

| Experiment number | w/o centralizing feature maps $X$ | Centralizing weights $W$ | FP statistics | BP statistics | Top-1 Error (%) |
|---|---|---|---|---|---|
| ① | ✓ | ✓ | EMAS | SMAS | 23.58(MABN) |
| ② | ✓ | ✓ | SMAS | SMAS | 26.63 |
| ③ | ✓ | ✓ | EMAS | EMAS | 24.83 |
| ④ | ✓ | ✓ | EMAS | BS | 27.03 |
| ⑤ | ✓ | ✓ | BS | BS | 29.68 |
| ⑥ | ✓ | | EMAS | SMAS | 25.45 |
| ⑦ | ✓ | | EMAS | BS | 29.57 |
| ⑧ | ✓ | | BS | BS | 32.95 |
| ⑨ | | | BS | BS | 35.22 |
| ⑩ | | ✓ | BS | BS | 34.27 |
| ⑪ | | ✓ | BS | BS | 23.35(regular) |

Table 4: **Further ablation study on ImageNet with ResNet-50.** The normalization batch size is 2 in all experiments. The buffer size ($m$) is 16 and momentum is $0.98$ when using SMA statistics, otherwise the momentum is $0.9$. BS means vanilla batch statistics.

## B.3 Experiments on COCO

**Implementation details.** We train the Mask-RCNN pipeline from scratch with MABN. We train the model on 8 GPUs, with 2 images per GPU. We train our model using COCO 2014 train and trainval35k dataset. We evaluate the model on COCO 2014 minival dataset. We set the momentum $\alpha = 0.98$ for all MABN layers. We report the standard COCO merics $AP^{bbox}$, $AP_{75}^{bbox}$, $AP_{50}^{bbox}$ for bounding box detection and $AP^{mask}$, $AP_{50}^{mask}$, $AP_{75}^{mask}$ for instance segmentation. Other basic settings follow He et al. (2017).

**MABN used on heads.** We build mask-rcnn baseline using a Feature Pyramid Network(FPN)(Lin et al., 2017a) backbone. The base model is ResNet-50. We train the models for $2\times$ iterations. We use 4conv1fc instead of 2fc as the box head. Both backbone and heads contain normalization layers. We replace all normalization layers in each experiments. While training models with MABN, we use batch statistics in normalization layers on head during first 10,000 iterations. Table 5 shows the result. The momentum are set to 0.98 in BRN and MABN.

|  | $AP^{bbox}$ | $AP_{50}^{bbox}$ | $AP_{75}^{bbox}$ | $AP^{mask}$ | $AP_{50}^{mask}$ | $AP_{75}^{mask}$ |
|---|---|---|---|---|---|---|
| BN | **32.38** | 50.44 | 35.47 | **29.07** | 47.68 | 30.75 |
| BRN | **34.07** | 52.66 | 37.12 | **30.98** | 50.03 | 32.93 |
| SyncBN | **36.81** | 56.23 | 40.08 | **33.11** | 53.46 | 35.28 |
| MABN | **36.50** | 55.79 | 40.17 | **32.69** | 52.78 | 34.71 |

Table 5: Comparision of Average Precision(AP) of Mask-RCNN on COCO Detection and Segmentation. The gradients batch size is 16. The normalization batch size of SyncBN is 16, while that of BN, BRN, MABN are both 2, the buffer size ($m$) of MABN is 32.

**Training from pretrained model.** We compare the performance of MABN and SyncBN when training model based on ImageNet pretrained weights for 2x iterations. The results are shown in Table

|  | $AP^{bbox}$ | $AP_{50}^{bbox}$ | $AP_{75}^{bbox}$ | $AP^{mask}$ | $AP_{50}^{mask}$ | $AP_{75}^{mask}$ |
|---|---|---|---|---|---|---|
| SyncBN | **38.25** | 57.81 | 42.01 | **34.22** | 54.97 | 36.34 |
| MABN | **38.42** | 58.19 | 41.99 | **34.12** | 55.10 | 36.12 |

Table 6: Comparision of Average Precision(AP) of Mask-RCNN on COCO Detection and Segmentation. The gradients batch size is 16. The normalization batch size of SyncBN is 16, while that of BN, BRN, MABN are both 2, the buffer size ($m$) of MABN is 32.

**Training from scratch for one-stage model.** We also compare MABN and SyncBN based on one-stage pipeline. We build on retinanet(Lin et al., 2017b) benchmark. We train the model from scratch for $2\times$ iterations. The results are shown in Table 7.

|  | $AP^{bbox}$ | $AP_{50}^{bbox}$ | $AP_{75}^{bbox}$ |
|---|---|---|---|
| SyncBN | **29.80** | 46.21 | 31.47 |
| MABN | **29.52** | 45.69 | 31.14 |

Table 7: Comparison of Average Precision(AP) of retinanet on COCO Detection. The gradients batch size is 16. The normalization batch size of SyncBN is 16, while that of MABN is 2.

All experiment results shows MABN can get comparable as SyncBN, and significantly outperform BN on COCO.

## B.4 Semantic Segmentation in Cityscapes

We evaluate semantic segmentation in Cityscapes(Cordts et al., 2016). It contains 5,000 high quality pixel-level finely annotated images collected from 50 cities in different seasons. We conduct experiments on PSPNET baseline and follow the basic settings mentioned in Zhao et al. (2017).

For fair comparison, our backbone network is ResNet-101 as in Chen et al. (2017). Since we centralize weights of convolutional kernel to use MABN, we have to re-pretrain our backbone model on Imagenet dataset. During fine-tuning process, we linearly increase the learning rate for 3 epoch (558 iterations) at first. Then we follow the "poly" learning schedule as Zhao et al. (2017). Table 8 shows the result of MABN compared with vanilla BN, BRN and SyncBN. The buffer size ($m$) of MABN is 16, the modementum of MABN and BRN is 0.98.

Since the statistics (mean and variance) is more stable in a pre-trained model than a random initialized one, the gap between vanilla BN and SyncBN is not significant (+1.41%). However, MABN

|  | pretrain Top-1 | mIoU |
|---|---|---|
| BN | 21.74 | 77.11 |
| BRN | 21.74 | 77.30 |
| SyncBN | 21.74 | 78.52 |
| MABN | 21.70 | 78.20 |

Table 8: Results on Cityscapes testing set.

still outperforms vanilla BN by a clear margin.(+1.09%). Besides, BRN shows no obvious superiority to vanilla BN(+0.19%) on Cityscapes dataset.

### B.5 COMPUTATIONAL OVERHEAD

We compare the computational overhead and memory footprint of BN, GN and MABN. We use maskrcnn with resnet50 and FPN as benchmark. We compute the theoretical FLOPS during inference and measure the inference speed when a single image ($3\times224\times224$) goes through the backbone (resnet50 + FPN). We assume BN and MABN can be absorbed in convolution layer during inference. GN can not be absorbed in convolution layer, so its FLOPS is larger than BN and MABN. Besides GN includes division and sqrt operation during inference, therefore it's much slower than BN and MABN during inference time.

We also monitor the training process of maskrcnn on COCO (8 GPUs, 2 images per GPU), and show its memory footprint and training speed. Notice we have not optimized the implementation of MABN, so its training speed is a little slower than BN and GN.

|  | FLOPS (M) | Memory (GB) | Training Speed (iter/s) | Inference Speed (iter/s) |
|---|---|---|---|---|
| BN | 3123.75 | 58.875 | 2.35 | 12.73 |
| GN | 3183.28 | 58.859 | 2.22 | 6.34 |
| MABN | 3123.75 | 60.609 | 1.81 | 12.73 |

Table 9: Computational overhead and memory footprint of BN, GN and MABN.

