# OpenReview forum: "Towards Stabilizing Batch Statistics in Backward Propagation of Batch Normalization"
_ICLR.cc/2020/Conference — Accept (Poster)_

### Official Review · AnonReviewer3 · 2019-10-22
**Official Blind Review #3**

**Rating:** 6

**Review:**

This paper provides a new method to deal with the small batch size problem of BN, called MABN. Compared to BRN, MABN has two main contributions: 1) find two statistics, g and ψ, in BP to apply moving average operation without introducing too much overhead; 2) reduce the number of statistics of BN via centralizing weight for better stability.

Though, I still have several concerns:
1.	You mentioned that “centralizing weights ... to satisfy the zero mean assumption” in Section 4.2. In other words, given E(W-mean(W))=0, E(X_{output})=E(W-mean(W))E(X_{input})=0. This equation holds only when W-mean(W) and X_{input} are irrelevant. However, model weights are updated based on the input and the loss function during training. Do you have any proof to ensure W-mean(W) and X_{input} are irrelevant?

2.	Your “Modified Structure” contains two operations, centralizing weights and reducing BN’s statistics. As much as I know, weight standardization can significantly improve BN’s performance, but the effect of weight centralization, i.e. “part” of weight standardization, is unclear yet. Therefore, I think a vanilla BN with weight centralization should also be included in ablation study. This experiment can help us better understand the pure effectiveness of reducing BN’s statistics.

3.	I think the results on COCO should be discussed in detail since object detection is an important task in computer vision and suffers from the small batch size regime. I wonder if it’s possible to compare MABN with SyncBN/GN on a higher baseline, such as finetuning from ImageNet pretrained model or training from scratch with at least 6x scheduler.

4.	What about the FLOPS, memory footprint and practical speed? These results would affect the application value of MABN. It would also be great if you can release the code, which would upgrade my rating a lot.

5.	What is the value of m for SMA? Is this value changed in different datasets? This would affect the performance of the proposed method.


Although there are several issues not addressed by the authors (the assumption of this paper  "W-mean(W) and X_{input} are irrelevant"; why not using ImageNet pre-trained model on COCO), I keep my initial rating of weak accept.

**Experience Assessment:**

I have published in this field for several years.

**Review Assessment: Checking Correctness Of Derivations And Theory:**

I assessed the sensibility of the derivations and theory.

**Review Assessment: Checking Correctness Of Experiments:**

I carefully checked the experiments.

**Review Assessment: Thoroughness In Paper Reading:**

I read the paper thoroughly.

---

> ### Author Response · Authors · 2019-11-13
> **Author Response to Official Blind Review #3**
>
> We’re thankful to the reviewer’s comments and conducted suggestions. We will response his/her concerns one by one.
>
> 1. “You mentioned that “centralizing weights ... to satisfy the zero mean assumption” in Section 4.2. In other words, given E(W-mean(W))=0, E(X_{output})=E(W-mean(W))E(X_{input})=0. This equation holds only when W-mean(W) and X_{input} are irrelevant. However, model weights are updated based on the input and the loss function during training. Do you have any proof to ensure W-mean(W) and X_{input} are irrelevant?”
>
> We admit modified normalization can worsen the performance of the model due to the lack of centralizing feature maps, and we have emphasized weight centralization (WC) is only a practical compensation without any theoretical guarantee. We clarify the effectiveness of WC by ablation study, the experiment results is shown in Table 4 in appendix. It can be seen WC indeed improve the performance of MABN with a clear margin (compare 1 and 6 in table 4). More accurate theoretical analysis remains as a future work.
>
> 2. “Your “Modified Structure” contains two operations, centralizing weights and reducing BN’s statistics. As much as I know, weight standardization can significantly improve BN’s performance, but the effect of weight centralization, i.e. “part” of weight standardization, is unclear yet. Therefore, I think a vanilla BN with weight centralization should also be included in ablation study. This experiment can help us better understand the pure effectiveness of reducing BN’s statistics.”
>
> We added the experiment of vanilla BN with WC in revised version (10, 11 in Table 4 in appendix B.2) as the reviewer’s request. In worst case (vanilla BN, small batch size), WC can improve the performance of vanilla BN from 35.22% to 34.27% with a clear margin, in best cases (vanilla BN, regular batch size), WC can only improve vanilla BN slightly, from 23.41% to 23.35%. While WC can significantly improve MABN even close to the best case, from 25.45% to 23.58%. We think WC can significantly improve the performance of MABN by compensating for the lack of centralizing feature maps.
>
> 3. “I think the results on COCO should be discussed in detail since object detection is an important task in computer vision and suffers from the small batch size regime. I wonder if it’s possible to compare MABN with SyncBN/GN on a higher baseline, such as finetuning from ImageNet pretrained model or training from scratch with at least 6x scheduler.”
>
> We are working on comparing MABN with SyncBN/GN in pretrained and 6x settings. However it’s more time-consuming than we thought, so there’s no guarantee we can add the experiment result by the due of rebuttal. But we will add the further experiment results in the final version if this work is accepted.
>
> 4. “What about the FLOPS, memory footprint and practical speed? These results would affect the application value of MABN. It would also be great if you can release the code, which would upgrade my rating a lot.”
>
> We added the results of FLOPS, memory footprint and practical speed in appendix B.5. Notice we haven’t done any computation optimization on the implementation of MABN, so MABN seems a little slow. We decide to release the code to verify the implementation of MABN, the dropbox link https://www.dropbox.com/sh/5jrhc6knovqecls/AAA5asCVvte945ulT5PtECA_a?dl=0 It includes the codes and checkpoint of MABN on Imagenet and COCO. The code is still messy, we are working on cleaning it up and releasing an official version to include all experiment code mentioned in the paper.
>
> 5. “What is the value of m for SMA? Is this value changed in different datasets? This would affect the performance of the proposed method.”
>
> In all experiment, the buffer size (m) of MABN is set as 16. The buffer size (m) depends on what the size of batch the user want MABN to simulate. We add the annotation in the revised version. We found using very large buffer size of MABN can’t improve the performance: on the one hand, the bias of SMAS of g and \phi rapidly increase as buffer size grows; on the other hand, it has been proven both in experiments and theoretical analysis that appropriate randomness of batch statistics indeed help improve the generalization performance of neural network[1, 2].
>
> [1] You Y, Gitman I, Ginsburg B. Large batch training of convolutional networks[J]. arXiv preprint arXiv:1708.03888, 2017.
> [2] Zhou A, Ma Y, Li Y, et al. Towards Improving Generalization of Deep Networks via Consistent Normalization[J]. arXiv preprint arXiv:1909.00182, 2019.

---

### Official Review · AnonReviewer1 · 2019-10-22
**Official Blind Review #1**

**Rating:** 8

**Review:**

The paper proposes a new approach for batch-normalization. Standard approaches are sensitive to the batch size, because small batches will lead to unstable statistics. So when the mini-batch is small, the performance can drop significantly. The paper addresses this issue by analyzing extra statistics in the batch normalization and introducing moving average statistics, weights centralization and a slightly modified normalization. The proposed method does not require large batch sizes and nonlinear operations, but still maintain the robustness. The theoretical analysis and guarantees are provided as well. Experiments on typical datasets demonstrate the effectiveness of the proposed trick.

Overall, the idea is interesting to me. The work is solid in both theory and practice. Hopefully the proposed scheme has the potential to fundamentally enhance the training of deep neural networks.

**Experience Assessment:**

I do not know much about this area.

**Review Assessment: Checking Correctness Of Derivations And Theory:**

I assessed the sensibility of the derivations and theory.

**Review Assessment: Checking Correctness Of Experiments:**

I assessed the sensibility of the experiments.

**Review Assessment: Thoroughness In Paper Reading:**

I read the paper at least twice and used my best judgement in assessing the paper.

---

> ### Author Response · Authors · 2019-11-13
> **Author Response to Official Blind Review #1**
>
> We really appreciate for the reviewer’s praise and encouragement. We are trying our best to refine our work and extend its usage in deep learning field.

---

### Official Review · AnonReviewer2 · 2019-10-23
**Official Blind Review #2**

**Rating:** 6

**Review:**

The paper extends recently proposed BatchRenormalization (BRN) technique which uses exponential moving average (EMA) statistics in forward and backward passes of BatchNorm (BN) instead of vanilla batch statistics. Motivation of the work is to stabilize training neural networks on small batch size setup. Authors propose to replace EMA in backward pass by simple moving average (SMA) and show that under some assumptions such replacement reduces variance. Also they consider slightly different way of normalization without centralizing features X, but centralizing convolutional kernels according to Qiao et al. (2019).

Concerns:
Changing batch statistics with moving averages in Eq. (4) and Eq. (16) may introduce bias in stochastic gradients. Authors do not reflect this problem in the work.
Assumption (3) from Theorem 3 does not hold in practice since authors centralize weights not features. Authors do not study the influence of this on the performance of the method.
Authors’ main motivation is to study small batch size experimental setup which is important in segmentation and detection, but they don’t include main competitor (BRN) in these experiments.

Overall, the proposed method has a lack novelty and thorough comparison against BRN. Therefore, I would suggest rejecting the current version.
--------------------------------------------------------
Update after author rebuttal

Thank you for your clarification. Below I provide an updated review for the paper.

The paper proposes the improvement of batch normalization techniques for the case of small batch size. Authors reveal new statistics used in gradient calculation of original BatchNorm and show their instability in case of small batch size. Thereafter they improve the method by reducing the number of statistics in backward pass resulting in more stability and performance increase. The authors provide a good experimental study on the influence of individual parts of the method. However, I still have concerns about the strictnesses of theorems assumptions and interpretability of their results in practice (i.e., o(1) biases in Theorem 1, o(1) assumptions in Theorem 2).

Overall, almost all of my concerns were justified. Therefore I increase my score to 6.

**Experience Assessment:**

I have published in this field for several years.

**Review Assessment: Checking Correctness Of Derivations And Theory:**

I assessed the sensibility of the derivations and theory.

**Review Assessment: Checking Correctness Of Experiments:**

I assessed the sensibility of the experiments.

**Review Assessment: Thoroughness In Paper Reading:**

I read the paper at least twice and used my best judgement in assessing the paper.

---

> ### Author Response · Authors · 2019-11-13
> **Author Response to Official Blind Review #2**
>
> We’re thankful to the reviewer for the comments and suggestions. The reviewer's main concern was on the lack novelty and thorough comparison against, which we address below one-by-one:
>
> Major concerns:
> 1. “the proposed method has a lack novelty”.
>
> First of all, we have to point out some misunderstanding about our work in reviewer’s comment: “Authors propose to replace EMA in backward pass by simple moving average (SMA) and show that under some assumptions such replacement reduces variance.” Combining with preceding context, the reviewer think MABN just replace SMAS of statistics associated with feature maps (mean, variance) used by BRN during backward passes. It’s totally wrong! MABN remains to use EMAS of feature maps as BRN, but replace extra batch statistics associated with gradients (g, \phi defined in equation (5) in the paper) of BN by SMAS during backward passes. We humbly suggest the reviewer to read our paper again.
>
> Our major contribution is to reveal the existence of batch statistics (g and \phi) associated with gradients from BN during backward propagation (BP), and prove they will severely affect the training process of neural network in small batch cases. To our best knowledge, the existence and influence of g and \phi in BN had never been mentioned or discussed by others even so far.
>
> Another major contribution is, we showed replacing batch statistics of g and \phi with SMAS make it possible to completely make up the defect of vanilla BN in small batch cases, while BRN has been widely known not useful to address small batch issues in practice, our experiments presented in last and revised version (table1, 3 in main body, table 4, 5, 7 in appendix) also proves it. Though MABN and BRN indeed have something in common, MABN solves the small batch problem in essence, while BRN do not remedy the primary problem.
>
> Besides, we do not purely take use of weight centralization (WC) as a modification of Weight Standardization (WS) to improve our method. Our real contribution is to modify the structure of normalization (from (x –mean)/std to x/E(x^2)) to reduce the number of batch statistics. We prove the effectiveness of modified normalization by both theoretical proof (Theorem 2 in section 4.1) and experiments (see 1, 6 in Table 2 in section 5.1). We admit modified normalization can worsen the performance due to lack of centralization. Thus we remedy the defect by centralizing weights. To clarify the effectiveness of WC, we added the ablation study on WC (see 10, 11, in Table 4 in appendix B.2), it can be shown WC don't significantly improve vanilla BN, but it can improve the performance of MABN with a clear margin (+1.87%).
>
> 2. “lack thorough comparison against BRN”, “Authors’main motivation is to study small batch size experimental setup which is important in segmentation and detection, but they don’t include main competitor (BRN) in these experiments.”
>
> BRN is not a main competitor of MABN on detection and segmentation tasks, for its bad performance. Our main competitor is SyncBN. For integrity, we added the experiments with BRN in detection and segmentation tasks (see Table 3, Table5, Table 7) in the revised version. All experiment results shows BRN can slightly improve the performance of vanilla BN in small batch cases, but still remains a huge gap from MABN and SyncBN.
>
> 3. “Changing batch statistics with moving averages in Eq. (4) and Eq. (16) may introduce bias in stochastic gradients. Authors do not reflect this problem in the work.”
>
> We have theoretically demonstrated the bias of moving averages statistics is extremely small compared with mean and variance by the equation (11) in Theorem 2, we also have illustrated the result by experiment, which is shown in Figure 4 in appendix B.1 in last version. We highlight the result in the main body in the revised version.
>
> 4. “Assumption (3) from Theorem 3 does not hold in practice since authors centralize weights not features. Authors do not study the influence of this on the performance of the method.”
>
> We think you mean theorem 2. Since we can’t strictly prove assumption (3) in theorem 2 can be satisfied by centralizing weights, we have pointed out WC is just a practical compensation, and carefully studied its influence by experiments, we presented the experiment result in Table 4 in last version. WC can improve the performance of MABN with a clear margin (1.87%), but without WC, MABN still significantly outperform its counterpart BRN by 4.84%. We highlight the result in the revised version.
>
> In addition, we have to point out some mistakes in reviewer’s comment: BRN is proposed in Feb. 2017. It has been almost 3 years since then, which is not a short time in deep learning fields, it seems not appropriate to call it “recently proposed techniques”, more importantly, BRN just use EMAS of mean and variance of feature maps as renormalize factors, but still use batch statistics (mean and variance) and their gradients during backward pass.

---

### Author Response · Authors · 2020-01-19
**The camera ready version has been released.**

Hi all,

We have uploaded the camera ready version of the paper, and the official code has been released. We added the results of mask-rcnn on COCO with pre-trained weights in the appendix.

Any question or discussion is welcome!

Best,
Ruosi Wan

---

### Decision · Program_Chairs · 2019-12-19

**Decision:**

Accept (Poster)

**Comment:**

This work introduces Moving Average Batch Normalization (MABN) method to address performance issues of batch normalization in small batch cases. The method is theoretically analyzed and empirically verified on ImageNet and COCO.
Some issues were raised by the reviewers, such as restrictive nature of some of the assumptions in the analysis as well as performance degradation due lack of centralizing feature maps. Nevertheless, all the reviewers found the contributions of this paper interesting and important, and they all recommended accept.